# Lectin-Based Approaches to Analyze the Role of Glycans and Their Clinical Application in Disease

**DOI:** 10.3390/ijms251810231

**Published:** 2024-09-23

**Authors:** Hiroko Ideo, Akiko Tsuchida, Yoshio Takada

**Affiliations:** Laboratory of Glycobiology, The Noguchi Institute, 1-9-7, Kaga, Itabashi, Tokyo 173-0003, Japan; akikots@noguchi.or.jp (A.T.); takada@noguchi.or.jp (Y.T.)

**Keywords:** lectin, glycosylation, glycosphingolipids, glycoprotein, cancer, analysis, galectin-4

## Abstract

Lectin-based approaches remain a valuable tool for analyzing glycosylation, especially when detecting cancer-related changes. Certain glycans function as platforms for cell communication, signal transduction, and adhesion. Therefore, the functions of glycans are important considerations for clinical aspects, such as cancer, infection, and immunity. Considering that the three-dimensional structure and multivalency of glycans are important factors for their function, their binding characteristics toward lectins provide vital information. Glycans and lectins are inextricably linked, and studies on lectins have also led to research on the roles of glycans. The applications of lectins are not limited to analysis but can also be used as drug delivery tools. Moreover, mammalian lectins are potential therapeutic targets because certain lectins change their expression in cancer, and lectin regulation subsequently regulates several molecules with glycans. Herein, we review lectin-based approaches for analyzing the role of glycans and their clinical applications in diseases, as well as our recent results.

## 1. Introduction

Given that glycans cover the outermost layer of the cell, various biological phenomena cannot be explained without deep insight into their functions. Although defects in proteins (genes) often have fatal consequences, those in glycans have less critical effects. The majority of attached glycans have little or no effect on protein function; however, certain glycans function in proper protein folding and act as platforms for processes such as communication, signal transduction, and adhesion. Congenital disorders of glycosylation (CDG, formerly called carbohydrate-deficient glycoprotein) are rare congenital disorders in which the glycosylation of various tissue proteins and/or lipids is either defective or absent. CDG syndromes cause serious malfunctions in several organ systems, especially the nervous system, muscles, and intestines [1]. Thus, glycans play an important role in organs affected by CDG syndrome.

Cancer, infection, and immunity are important clinical aspects of glycan function. Hence, lectins play an important role in these processes as glycan receptors (Figure 1). Proteins are assembled from amino acids using the information encoded by genes, whereas glycans are assembled from monosaccharides using various glycosyltransferases. Glycans conjugated to proteins and lipids confer specific structures and functions. Unlike amino acids and nucleotides, two monosaccharides can bind to several binding sites, resulting in several isomeric structures. In addition, one monosaccharide can form glycosidic linkages at multiple sites, thereby exponentially increasing the number of possible structures. Accordingly, the recognition of glycans by lectins is not simple, and the three-dimensional structure and multivalency of glycans and their components are important. Because glycans and lectins are inextricably linked, studies on lectins have also led to research on the roles of glycans. Compared with the advances in gene and protein analyses, advances in glycan analysis have been slower owing to the complexity of glycans. However, recent advances in mass spectrometry (MS) and related technologies have allowed for progress in this field [2].

The types of lectins and their functions have been extensively reviewed [3]; therefore, this review focused on aspects related to diseases and lectins while providing specific research examples.

## 2. Lectins as a Tool for Analyzing Glycosylation

The discovery of molecules that cause red blood cells to aggregate led to the discovery of molecules that recognize carbohydrates, namely lectins. Purified (plant) lectins are an important tool because they are used for blood group antigen typing. Moreover, lectins have long been used as an important tool for glycan structural analysis after elucidating their distinct carbohydrate specificities. Lectins exist not only in plants, but also in animals, fungi, viruses, and many other organisms. Antibodies that recognize glycans and/or glycopeptides can also be considered lectins in a broader sense. The binding specificities of various lectins and analytical methods using lectins have been extensively explored [4].

There are various analytical methods that use lectins, including the use of lectin-affinity chromatography, lectin microscopy, lectin-overlay enzyme-linked immunosorbent assay (ELISA), lectin blotting, and lectin microarrays. MS provides information on the type and amount of glycans released from glycoproteins and glycosphingolipids (GSLs); however, in most cases, spatial information is lost. Therefore, lectin-based analysis remains useful in many situations, and new technologies are emerging. A clinical test of the lectin electrophoresis system was established to identify increased fucosylated alpha-fetoprotein levels in liver cancer [5]. Lectin-based glycan profiling of single cells using sequencing (scGlycan-seq) is a glycan-profiling technology that uses multiple DNA-barcoded lectins and a next-generation sequencer (DNA-decoding device) [6,7]. More precise information can be obtained by combining lectin-based immunohistochemical analysis and MS imaging techniques rather than by applying individual analyses [8,9].

Recent remarkable improvements in the performance of mass spectrometers have made them important tools for glycan analysis. However, because of the high cost of mass spectrometers, they are not easily accessible. Moreover, understanding MS principles and correctly interpreting mass spectra when analyzing glycans is difficult. Given that the three-dimensional structure and multivalency of glycans are important factors for their function, the binding character to lectins provides important information.

### 2.1. Lectins as a Tool for Detecting Cancer-Related Changes in Glycosylation

Cancer cells acquire abnormal cell properties, and changes in cell-surface glycoconjugates occur simultaneously. Therefore, various cancer-associated glycan modifications, including fucosylation and sialylation, have been used for cancer diagnosis [10]. A high degree of N-linked glycan branching is frequently observed in glycoproteins of cancer cell origin, especially in those with metastatic potential [11]. In the absence of MS analysis, labeled glycans are subjected to gel filtration or electrophoresis to determine their molecular weight, and terminal structures are determined by a combination of lectin chromatography and glycosidase digestion. This method remains useful with some modifications. Hereafter, we provide specific examples of glycan analysis using lectins.

#### 2.1.1. Lectin Chromatography

Prostate-specific antigens (PSAs) are the most frequently used biomarker in prostate cancer screening. We explored new candidates for cancer-related glycoforms of PSAs to solve the problem of the high false-positive rate of PSA testing [12]. Xenografts formed from cancer tissue-originated spheroids (CTOS) resemble the key features of parental tumors [13]. Therefore, CTOS-derived PSAs reflect the glycan structure of a patient’s tumor. We found that PSA molecules with high and low molecular weights, which passed through a concanavalin A (Con A) column (Con A (−) fraction), were secreted from both CTOS and other cancer cells and were almost negligible (below 2%) in seminal PSAs from healthy men (Figure 2A). Further analysis revealed that the Con A (−) fraction of PSAs contained multi-antennary complex-type glycans (Figure 2B) or was non-glycosylated. PSA glycoforms were analyzed using a combination of *Wisteria floribunda* agglutinin (WFA), Con A, *Datura stramonium* agglutinin, and galectin-1 lectins and were confirmed using MS analysis [12]. Moreover, Haga et al. [14] observed hyperbranching of PSA glycans secreted into the serum of patients with cancer using liquid chromatography–tandem MS. However, the non-glycosylated form may not be detected without a lectin-based approach. We also found the non-glycosylated form of transferrin in patients with CDG using lectin chromatography [15]. Many studies that analyze cancer-related changes in glycans after their release may miss the non-glycosylated form. This lectin-based method is advantageous because it detects cancerous changes of glycans in a small amount of glycoproteins before (without) the use of MS equipment.

#### 2.1.2. Lectin-ELISA

There are several examples of the application of lectin-ELISA to detect glycosylation changes using clinical samples [16,17,18]. Membrane-associated mucin-type 1 (MUC1) is a heavily O-glycosylated transmembrane protein localized in normal secretory epithelial cells. The aberrant expression and glycosylation of MUC1 have been demonstrated in various cancers; accordingly, MUC1 is frequently used as a therapeutic marker in clinical applications [19]. Moreover, MUC1 is involved in cancer invasion, metastasis, angiogenesis, and the regulation of related biomolecules. Some MUC1 assays focus on changes in glycosylation and not only the amount of MUC1. For example, truncated O-glycans including GalNAcα- (Tn), Neu5Acα2,6-Tn (sTn) Galβ1,3-GalNAcα- (TF), Neu5Acα2,6-TF, and Neu5Acα2,3-TF are found in breast, prostate, colon, respiratory, pancreas, ovarian, and gastric cancers. In addition, sialyl-Lewis antigens, including NeuAcα2,3-Galβ1,3-(Fucα1,4)-GlcNAc-R (sLe^a^,) and NeuAcα2,3-Galβ1,3-(Fucα1,3)-GlcNAc-R (sLe^x^) are also found. Furthermore, WFA-positive sialylated MUC1 is a sensitive biomarker for biliary tract carcinoma and intrahepatic cholangiocarcinoma [20,21]. Additionally, plant lectins have actively been used to detect changes in MUC1 glycans; however, galectin-4, a mammalian lectin, can also be used to detect MUC1 glycans.

The cancer antigen 15-3 assay (CA15-3) has been widely used for the detection of breast cancer recurrence, and the epitope detected in the CA15-3 assay is an extracellular domain of MUC1. However, the assay’s sensitivity and specificity are inadequate. The breast cancer cell line YMBS secretes sulfated mucin 1 including 3′-sulfated Galb1-3GalNAc [22]. Therefore, we evaluated whether sulfated MUC1 was secreted into the blood of patients with breast cancer [23]. PNA, a Galb1-3GalNAc (core 1)-specific lectin, scarcely bound to MUC1 in YMBS; however, its binding to PNA greatly increased after sialidase treatment. Similarly, MUC1 captured from the sera of patients with breast cancer scarcely bound to PNA, suggesting that most of core 1 was modified with sialic acid or sulfate. We previously found that galectin-4 preferentially recognized the 3-O-sulfated core1 structure [24], and we developed a lectin-sandwich immunoassay, called Gal4/MUC1, using galectin-4 and a MUC1 monoclonal antibody (Figure 3A). The Gal4/MUC1 assay detected MUC1 in patient blood without sialidase treatment (Figure 3B). Receiver operating characteristic curve analysis was performed using the values from the Gal4/MUC1 and CA15-3 assays on sera from healthy individuals and patients with relapsed/metastatic breast cancer (Figure 3C). The positivity ratio in the Gal4/MUC1 assay was higher than that in the CA15-3 assay, particularly in relapsed or metastatic breast cancer. These results suggested that animal-derived lectins are effective alternatives for detecting cancerous glycans.

#### 2.1.3. Lectin Microarray

A lectin microarray is a technology in which a variety of lectins and antibodies are immobilized on a slide for the high-throughput analysis of glycans and glycoproteins [25,26]. Lectin microarray technology is useful for the high-throughput glycan analysis of complex biological samples. This approach is particularly effective for determining whether biological samples differ in glycosylation. There are several types of lectin microarrays, including direct assays, lectin-overlay antibody sandwich arrays, and antibody-overlay lectin sandwich arrays (Figure 4). In the direct assay, fluorescently labeled samples are added to slides on which lectins with distinct carbohydrate-binding specificities are immobilized, and the fluorescence patterns are compared to analyze the differential glycosylation patterns of normal versus diseased samples [27]. Lectin-overlay antibody sandwich arrays use antibody microarrays to capture multiple proteins and biotinylated lectin probes. Cancer-associated changes in the glycans of MUC1 and carcinoembryonic antigens were identified using this array [25,26]. Because this technique does not allow for a complete determination of glycan structures and is not quantitative, it is more appropriate for comparative studies, such as the comparison of glycan profiles.

## 3. Lectins as a Receptor for Glycans

### 3.1. Glycosphingolipids and Lectins in Cancer

GSLs are a group of lipids composed of hydrophobic ceramides and carbohydrates that are a part of the cell membrane. These carbohydrates on the cell surface play important roles in processes such as cell adhesion, cell–cell interactions, infection, and oncogenesis. Malignant transformation during cancer progression is often associated with changes in GSL glycan structure, and these cancer-associated GSLs have been used as diagnostic markers and targets for immunotherapy [28]. Recent advances in analytical and genome-editing technologies have also demonstrated the role of GSLs as modulators of immune cell function [29]. GSL glycans can be analyzed by MS after endoglycoceramidase release and using lectins, including monoclonal antibodies that target specific glycans (Le^x^, Le^a^, and GD2) and toxins (cholera toxin for GM1). Lectin arrays can also be used to analyze GSL glycans [30]. GSLs serve as signaling platforms in lipid rafts, and their glycans influence the malignant potential of cancers.

#### 3.1.1. Gangliosides

Ganglioside expression positively or negatively regulates several types of signaling molecules, including the epidermal growth factor receptor and mesenchymal-epithelial transition factor, in malignant cancer cells [31]. Disialyl gangliosides (GD3 and GD2) enhance the malignant properties of cancer cells, whereas monosialyl gangliosides (GM3, GM2, and GM1) exert the opposite effects [32,33]. Therefore, the development of a therapeutic strategy targeting cancer-specific gangliosides may be valuable in cancer treatment [34]. The endogenous sialic acid-binding lectins in humans belong to the Siglec family, most of which are expressed on immune cells. In the nervous system, Siglec-4 binds to gangliosides GD1a and GT1b, and Siglec-7 on natural killer cells binds to gangliosides GD3 and GD2 to inhibit immune signaling. The expression of GD3 and GD2 in cancer cells can lead to tumor immune evasion. Siglec-1 on macrophages binds to gangliosides on tumors, thereby enhancing antigen presentation. Gangliosides are found in human cells and tissues in a cell-specific distribution and are functional Siglec ligands with varied roles [34]. 

#### 3.1.2. GSLs with Blood Group Antigen

In addition to gangliosides, the expression of neutral GSLs, especially blood group antigens, is altered during tumor formation. The Sd^a^ blood group carbohydrate structure and the glycosyltransferase responsible for Sd^a^ synthesis (Sd^a^-β1,4GalNAcT) activity were remarkably decreased in cancer lesions of the gastrointestinal tract [35]. Moreover, the forced expression of Sd^a^ causes a remarkable deduction of carbohydrate ligands for selectin and increases the metastatic potential of human gastrointestinal tract cancer cells [35]. Furthermore, GSL suppression in certain blood group carbohydrate structures (Le^a^, Le^b^, H type-1, -2, A type-1, and-2) has been reported in gastric cancer specimens [36].

#### 3.1.3. GSLs with β1,3-Linked Galactose

We observed that galectin-4 was highly expressed in poorly differentiated gastric cancer cells with high metastatic potential and that galectin-4 suppression notably impeded peritoneal metastasis in murine models (Figure 5A) [37]. A detailed glycan analysis was performed by comparing highly metastatic wild-type and low metastatic galectin-4 knockout (KO) cells [38]. NUGC4 wild-type and KO cells mainly had highly fucosylated, highly branched galactose termini and bisected GlcNAc N-glycan structures; however, there were no significant differences between them. The further structural analysis of GSL glycans using glycosidase and MS^n^ analysis showed that GSL glycans with β1,4-linked galactose at the non-reducing termini, including Le^X^, were present in the wild type. In addition, GSL glycans with β1,3-linked galactose, including Le^a^ and Type 1A, were observed in the KO cells (Figure 5B) [38]. In subsequent real-time polymerase chain reaction analyses, beta-1,3-galactosyltransferase 5 (B3GALT5) was identified as a candidate glycosyltransferase, which was inferred from the GSL biosynthetic pathway (Figure 6A,B), and stable B3GALT5-expressing clones were isolated [39]. The glycan profiles of GSLs from the high B3GALT5-expressing clones resembled those of KO cells, and there was a notable suppression of peritoneal metastasis in the mouse model (Figure 5C). Glycans presented on GSLs may affect the malignant properties of poorly differentiated gastric cancer cells. In this case, GSLs with β1,3-linked galactose, including blood group antigens, suppressed the metastasis of cancer cells.

## 4. Glycans as a Target of Lectins

### 4.1. Glycans as a Target of Pathogens

GSLs serve as receptors in pathogen invasion. Several pathogens such as cholera, tetanus, and botulinum use GSLs on the surface of host cells as binding receptors, and lipid rafts composed of GSLs may act as a platform for signaling in the presence of pathogens [40]. Furthermore, many pathogens recognize blood group antigens, and the glycosylation profile of the gastrointestinal tract leads to differences in the susceptibility to infectious and chronic diseases [41]. An association between blood group A and susceptibility to severe acute respiratory syndrome coronavirus 2 (SARS-CoV-2) infection was identified [42]. The receptor-binding domain of SARS-CoV-2 shares binding specificity toward blood type 1 A glycan with human galectin-4 and -8 and preferentially infects blood group A-expressing cells, thereby providing a direct link between blood group A expression and SARS-CoV-2 infection [43]. The pretreatment of blood group A-presenting cells with a blood group-binding galectin (galectin-4) specifically decreased SARS-CoV-2 infection, which was increased by blood group A presentation. Galectin-4 and -8, which are expressed in the intestinal tract, recognize and kill blood group antigen-expressing *Escherichia coli* [44].

An agricultural example of the application of bacterial lectin is *Bacillus thuringiensis* (Bt), which is the most widely used microbial biopesticide in agriculture. Bt is characterized by the production of endotoxin proteins, known as Cry proteins, which bind to receptors (glycoproteins or glycolipids) located in insect midgut epithelial cells. Bt Cry toxins are widely used in genetically modified crops because the differences in glycans between insects and mammals make them host-specific and safe biopesticides. Certain Cry proteins have been studied for use as anticancer agents [45]. Glycolipid-binding galectins function in host defense against Cry toxin-expressing bacterial infections by targeting glycolipids in *Caenorhabditis elegans* [46].

### 4.2. Lectin-Mediated Drug Delivery

The applications of lectins are not limited to glycan analysis. Cancer-associated modifications of glycans resulted in various forms of lectin-mediated drug targeting [47]. *Burkholderia cenocepacia* is a gram-negative bacterium and an opportunistic pathogen that infects patients, especially those with cystic fibrosis [48]. The recombinant N-terminal domain of BC2L-C (rBC2LCN) lectin identified from this bacterium recognizes tumor-associated glycan, H-type3, which is expressed on human pluripotent stem cells [48]. The rBC2LCN-conjugated bacterial exotoxin showed good results as a lectin drug conjugate in a mouse model and is a promising candidate for the diagnosis and treatment of pancreatic ductal adenocarcinoma [49,50].

## 5. Lectins as the Drug Target

In addition to using external lectins as tools, endogenous lectins can also serve as therapeutic targets. Certain lectins regulate several molecules involved in adhesion, apoptosis, immune response, and signal transduction and alter their expression in cancer, rendering lectins potential therapeutic targets. Many therapeutic studies have targeted mammalian lectins and their inhibitors, several of which are under development. Here are examples of therapeutic lectin targets.

### 5.1. Asialoglycoprotein Receptor

The asialoglycoprotein receptor (ASGPR), also known as the hepatic-binding protein, is the first Ca^2+^-dependent animal lectin that binds to galactosyl- and N-acetylgalactosaminyl-containing proteins. ASGPR removes these molecules from the circulation by transporting them to the liver. After the ligand binds to ASGPR, this complex is internalized and transported into hepatocyte endosomes, where the receptor and ligand are disassociated. ASGPR then returns to the cell membrane surface. This property of the ASGPR has led to the development of N-acetylgalactosamine (GalNAc) small-interfering RNA (siRNA) conjugates for delivery to the liver [51]. A liver-targeted siRNA-conjugated tri-antennary GalNAc greatly improves cell uptake because the multivalency of the sugar ligand enhances affinity toward the lectin receptor [52].

### 5.2. Selectins and Siglecs

Selectins are transmembrane proteins with Ca^2+^-dependent glycan-binding domains at the amino terminus, followed by a consensus epidermal growth factor-like domain. Three members of the selectin family show distinct expression patterns: E-selectin (in endothelial cells), L-selectin (in leukocytes), and P-selectin (in platelets). Selectins mediate immune cell adhesion to the endothelium to facilitate physiological responses, such as inflammation, immunity, and hemostasis [53]. Selectins promote various interactions between tumor cells and blood constituents, including platelets, endothelial cells, and leukocytes, during cancer progression. The common glycan motifs of the three selectins are the terminal sialyl-Lewis^x^ (sLe^x^) and its isomer sialyl-Lewis^a^ (sLe^a^). Considering that sLe^x^ and sLe^a^ are correlated with poor prognosis due to the enhanced metastatic phenotype in many tumor cells [54], selectins have been studied as therapeutic target molecules [53]. Many small molecule inhibitors and monoclonal antibodies against selectins have been used in clinical studies. Crizanlizumab, a humanized monoclonal antibody that binds to P-selectin, has been approved by the Food and Drug Administration (FDA) [55].

Siglecs are a family of receptors expressed by most immune cells that bind proteins and lipids in glycans containing sialic acid. There are 15 different mammalian Siglecs that provide different functions based on cell surface receptor–ligand interactions [34]. Many types of tumor cells are hyper-sialylated, generating ligands for Siglec receptors on the immune cells. Sialylated CD43, LGALS3BP, and CD24 have been identified as specific ligands for Siglec-7, Siglec-9, and Sglec-10, respectively [56]. The distinctive expression of Siglecs enables them to act as cell markers and has enabled therapies targeting Siglec-expressing cells. For instance, an antibody–drug conjugate (ADC) monoclonal antibody against Siglec-2 and Siglec-3 has received FDA approval, and several agents against various Siglecs are in preclinical studies [57]. Siglec-based therapeutics also include new therapeutic approaches, such as anti-Siglec bispecific T-cell engagers, and chimeric antigen receptor T-cell therapies [57]. Trans Siglec–sialic acid signaling can occur via cell-cell interactions, and cis Siglec–sialic acid signaling occurs via the binding of Siglecs on immune cells expressing sialic acid. Siglecs have been studied as therapeutic targets in cancer and drugs, including small molecules that inhibit their binding to ligands, ADC, monoclonal antibodies, etc. [55].

### 5.3. Galectins

Galectins are a family of soluble lectins that recognize glycans with β-galactoside, and approximately 15 galectins have been discovered in mammals. Galectins contain one or two carbohydrate-binding domains (CRD) per molecule with similar amino acid sequences and structures; however, each galectin has different sugar-binding specificity [58]. Galectins and glycan interactions are complex, and their multivalency and oligomeric state, as well as the multivalency of their ligands, affect their high-affinity binding. The functions of galectins include early development, tissue regeneration, cancer, immune homeostasis, and recognition/effector functions against potential pathogens [59]. Galectins have been reported to change their expression in response to cancer development in various tissues [60]. Moreover, galectins have been studied as therapeutic targets in cancer, particularly small-molecule inhibitors that target galectin-1, -3 tumor progression, and fibrotic diseases [61,62].

Although galectin-1 and -3 are ubiquitously expressed, galectin-4 is normally expressed in gastrointestinal epithelial cells. Galectin-4 has two CRDs with distinct binding specificities and functions that crosslink molecules and regulate several biological processes [63]. We recently found that galectin-4 participates in the peritoneal metastasis of malignant gastric cancer cells by interacting with several molecules, including c-MET and CD44, and affecting glycosylation [37,38,39].

The emergence of drug resistance limits the efficacy of targeted therapies for human tumors because most targeted therapies target a single molecule [64]. Lectins regulate many glycosylated molecules; therefore, they have high potential to control many related molecules as master regulators and are promising therapeutic targets (Figure 7).

Several types of anti-cancer therapy are in use, including chemotherapy, hormone therapy, and molecular-targeted therapy. Traditional chemotherapy uses one or more anticancer drugs which are nonspecific intracellular poisons that inhibit mitosis or damage DNA. Molecular target drugs are more specific than chemotherapy, but because most target molecules (such as tyrosine kinases) are important in normal cells, side effects cannot be ignored. In contrast, as shown in Figure 1, lectins play a role in biological defense, control the localization of various molecules, and regulate cell–cell and cell–matrix interactions. In general, lectin deficiencies do not show a serious phenotype, partly because of the redundancy of lectins. Lectins regulate the localization of binding molecules, controlling cell–cell and cell–matrix interactions, so as to not destroy the function of their binding molecules. In addition, sugar chains or compounds mimicking sugar chains often have low toxicity as inhibitors [61].

## 6. Conclusions

Spatial information on glycans, including multivalency, is an important factor in interpreting the role of glycans; therefore, lectin-based approaches for the analysis of glycans remain valuable. Glycans are present in the outermost layer of organisms and play an important role in cancer and infection. The spatial arrangement of glycans is of great significance for GSLs and their recognized proteins. Moreover, lectins are promising therapeutic targets because they regulate several molecules through glycans.

## Figures and Tables

**Figure 1 ijms-25-10231-f001:**
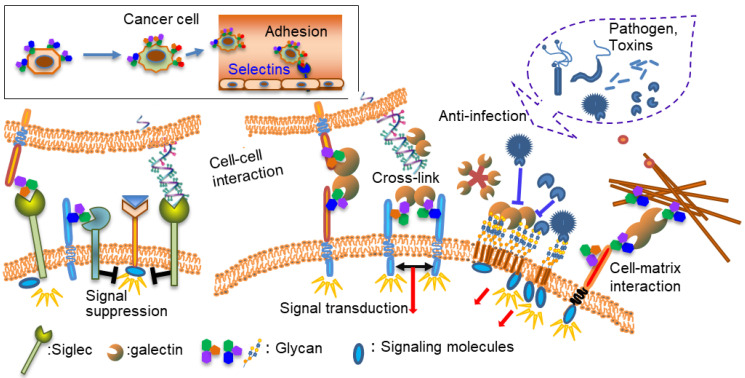
Functions of lectins and glycans.

**Figure 2 ijms-25-10231-f002:**
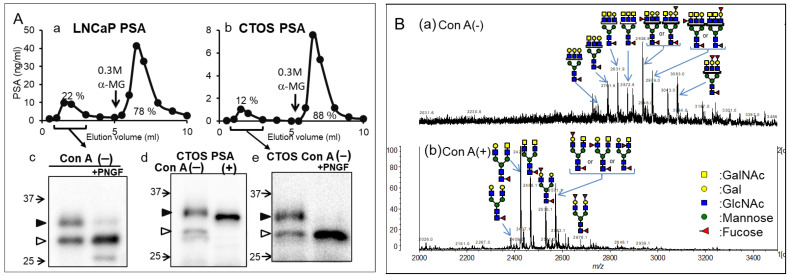
Concanavalin A (Con A)-unbound glycoforms in prostate-specific antigens (PSAs) from cancer cells. (**A**) Elution profiles of PSAs from cancer cells using Con A column chromatography. (**a**) LNCaP. (**b**) CTOS. Black arrows indicate the positions where the buffers were switched to those containing 0.3 M α-methyl glycoside (α-MG). Western blot analysis of PSAs from cancer cells. (**c**) PSAs from LNCaP in Con A (−) fraction with and without Peptide/N-glycosidase F (PNGF) treatment. (**d**) PSAs in Con A (−) and bound (+) fraction from CTOS. (**e**) PSAs in Con A (−) fraction from CTOS with and without PNGF treatment. The position of each high molecular and low molecular PSA is indicated by closed and open triangles, respectively. Following PNGF treatment, both high molecular weight forms (32 and 31 kDa) changed to the low molecular weight form (29 kDa). The majority of the 29-kDa form (open triangle) in Con A (−) fraction did not change, suggesting that it was non-glycosylated. (**B**) Analysis of PSA glycopeptides from LNCaP using matrix-assisted laser desorption ionization–mass spectrometry (MALDI-MS). MALDI-TOF MS spectra of glycopeptides in Con A (−) (**a**) and Con A (+) (**b**) fractions. Mass spectra were acquired in negative ion mode. (**A**,**B**) are reproduced from our previous study [12].

**Figure 3 ijms-25-10231-f003:**
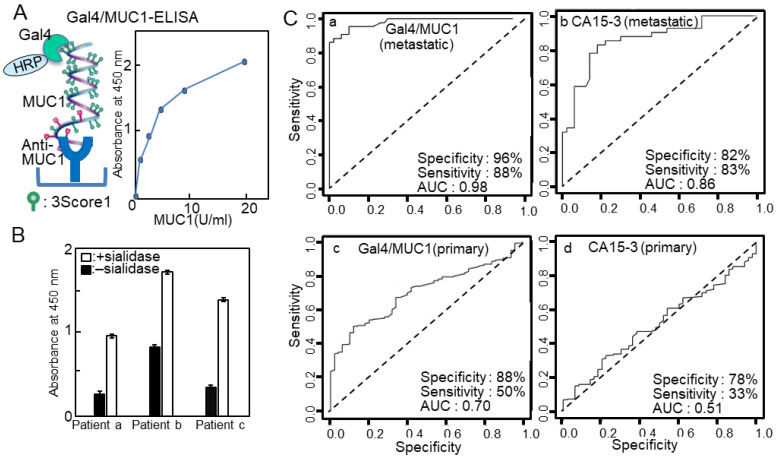
(**A**) Schema of galectin-4 (Gal4)/mucin-type 1 (MUC1)-enzyme-linked immunosorbent assay (ELISA) and standard curve. (**B**) Gal4/MUC1-ELISA of intact (solid bars) and sialidase-treated sera (white bars) from patients with breast cancer. Solutions of 1% sera were applied to anti-MUC1-coated plates and detected using Gal4-horseradish peroxidase (HRP). Background values, which were obtained from plates with blocking reagents only, were subtracted to account for nonspecific binding. (**C**) Receiver operating characteristic (ROC) plots displaying the specificities and sensitivities of (**a**) Gal4/MUC1 and (**b**) cancer antigen 15-3 (CA15-3) from patients with breast cancer with recurrence/metastasis, and the specificities and sensitivities of (**c**) Gal4/MUC1 and (**d**) CA15-3 from patients with primary breast cancer. (**A**,**C**) are reproduced from our previous study [23].

**Figure 4 ijms-25-10231-f004:**
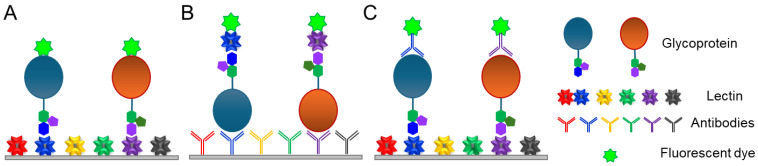
Lectin microarray strategies. (**A**) Direct assay. (**B**) Lectin-overlay antibody sandwich array. (**C**) Antibody-overlay lectin sandwich array. (**A**–**C**) are produced referring to the figure in the reference [27].

**Figure 5 ijms-25-10231-f005:**
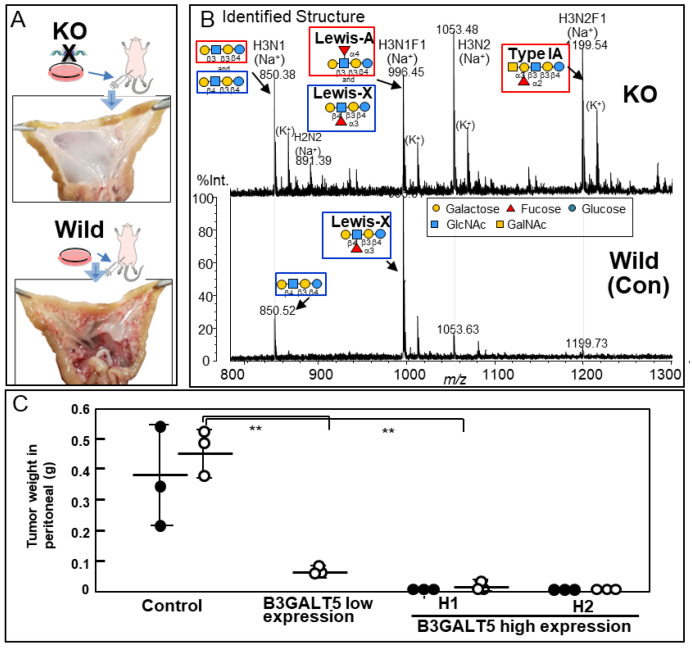
(**A**) Analysis of galectin-4 knockout (KO) cells in vivo. Macroscopic view of mesentery from mice inoculated with wild-type and KO cells. (**B**) The structures of the neutral GSL glycans from wild-type and galectin-4KO NUGC4 cells. The compositions and identified structures of the GSL glycans were acquired in positive ion mode. The glycan structures were identified using glycosidase digestion and multi-stage mass spectrometry (MS^n^) analysis. (**C**) Total weight of tumors in the peritoneal cavity in two animal experiments. The first experiment (black circles) and the second experiment (blank circles). The horizontal line in the middle of columns shows the average tumor weight of mice inoculated with NUGC4 clone cells (Control, B3GALT5 low expression, high expression H1 and H2). The accompanying vertical line indicates standard deviation (SD). ** *p* < 0.005 (**A**,**C**) are reproduced from our previous study [37,38,39].

**Figure 6 ijms-25-10231-f006:**
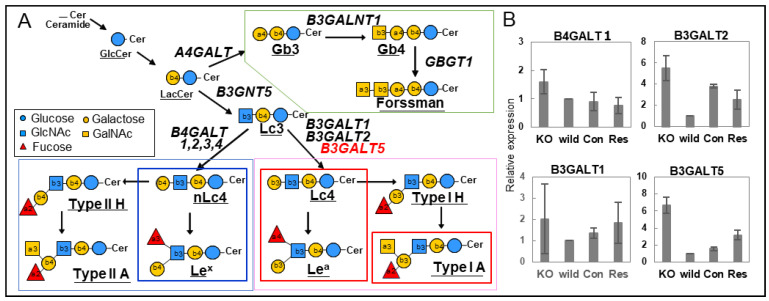
(**A**) Possible GSL biosynthetic pathways in NUGC4 cells. The synthesis of the three series of mammalian GSLs with the possible gene names. (**B**) Relative expression levels of beta-1,3-galactosyltransferase (B3GALT)1, 2, and 5 genes in the synthesis of the GSLs using quantitative real-time polymerase chain reaction (PCR). The normalized expression ratio of each gene against the glyceraldehyde 3-phosphate dehydrogenase (GAPDH) gene is represented on the vertical axis. (**A**,**B**) are reproduced from our previous study [39].

**Figure 7 ijms-25-10231-f007:**
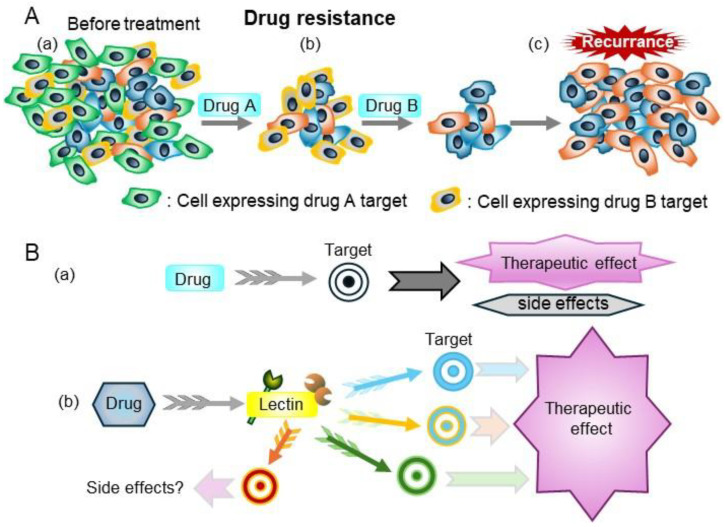
(**A**) Drug resistance is a major barrier to successful treatment. (**a**) Before treatment, tumors consist of various cancer cells with different molecular features. (**b**) Drug A and B kill some cancer cells that express the drug targets. (**c**) The cancer cells that are resistant to drugs grow, thereby contributing to re-growth of the tumor. (**B**) (**a**) Treatments that target specific molecules, such as molecular-target drugs and monoclonal antibodies. (**b**) Drugs that target lectins can regulate many molecules with specific glycans which lectin recognizes, thereby modulating cell properties.

## Data Availability

No new data were created or analyzed in this review.

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
