# Peer review of "Lectin-Based Approaches to Analyze the Role of Glycans and Their Clinical Application in Disease"

_ijms, 2024, doi:10.3390/ijms251810231_

Round 1
Reviewer 1 Report
Comments and Suggestions for Authors
See attached file

In general, the English is ok, BUT obviously the editing was done by someone who is not involved in science. Therefore, some construction of sentences are grammatically fine, but wrong or misleading regarding the content.
Reviewer 2 Report
Comments and Suggestions for Authors
In this valuable review by Ideo, tsuchida, and Takada, the various lectin-based approaches for analyzing the role of glycans and their clinical applications in diseases are reported. The topic of the review is not highly original, but as a matter of fact it is of quite interest for the community. The English language is of quality, the work comprehensively described, and well organized. Furthermore, the conclusions are consistent with the arguments presented. After careful reading I thus recommend publications in the Int. J. Mol. Sci. after the following minor revisions have been performed.
L13: is it of glycans or on glycans that the authors mean?
Fig.2B: the "little" drawings of polysaccharides are too small and hence non-conclusive.
L116-L125: the caption must be shifted at page 3.
L255-L263: the caption must be shifted at page 7.
Fig. 6 is of low quality. Please insert a high-quality illustration (PNG, JPEG, or TIFF).
Reference section: The good template for references must be used (correct abbreviated names have not been used for each journals).
